# The role of partner influence in contraceptive adoption, discontinuation, and switching in a nationally representative cohort of Ugandan women

**Dana O. Sarnak** [1]*, **Shannon N. Wood**[1], **Linnea A. Zimmerman**[1], **Celia Karp**[1], **Fredrick Makumbi**[2], **Simon P. S. Kibira**[3], **Caroline Moreau**[1,4]

**1** Department of Population Family and Reproductive Health, Johns Hopkins Bloomberg School of Public Health, Baltimore, Maryland, United States of America, **2** Department of Epidemiology and Biostatistics, School of Public Health, Makerere University, Kampala, Uganda, **3** Department of Community Health and Behavioural Sciences, School of Public Health, Makerere University, Kampala, Uganda, **4** Soins et Santé Primaire, CESP Centre for Research in Epidemiology and Population Health U1018, Inserm, Villejuif, France

* dsarnak1@jhu.edu

**Data Availability Statement:** Data for this study are publicly available at https://www.pmdata.org/data/available-datasets. Data are free to download

## Abstract

### Background

Understanding contraceptive use dynamics is critical to addressing unmet need for contraception. Despite evidence that male partners may influence contraceptive decision-making, few studies have prospectively examined the supportive ways that men influence women's contraceptive use and continuation.

### Objective

This study sought to understand the predictive effect of partner influence, defined as partner's fertility intentions and support for contraception, and discussions about avoiding pregnancy prior to contraceptive use, on contraceptive use dynamics (continuation, discontinuation, switching, adoption) over a one-year period.

### Methods

This study uses nationally representative longitudinal data of Ugandan women aged 15–49 collected in 2018–2019 (n = 4,288 women baseline; n = 2,755 women one-year follow-up). Two analytic sub-samples of women in union and in need of contraception at baseline were used (n = 618 contraceptive users at baseline for discontinuation/switching analysis; n = 687 contraceptive non-users at baseline for adoption analysis). Primary dependent variables encompassed contraceptive use dynamics (continuation, discontinuation, switching, and adoption); three independent variables assessed partner influence. For each sub-sample, bivariate associations explored differences in sociodemographic and partner influences by contraceptive dynamics. Multinomial regression models were used to examine discontinuation and switching for contraceptive users at baseline; logistic regression identified predictors of contraceptive adoption among non-users at baseline.

and use; users are required to register for a PMA dataset account. The specific datasets used in this study are the Uganda Round 6 (2018) Household & Female survey and the Uganda Round 6 Follow-up (2019) Household & Female survey.

**Funding:** This work was supported by the Bill & Melinda Gates Foundation, Seattle, WA [OPP1163880; Investigator: AO Tsui]. The funders had no role in study design, data collection and analysis, decision to publish, or preparation of the manuscript.

**Competing interests:** The authors have declared that no competing interests exist.

## Results

Among users at baseline, 26.3% of women switched methods and 31.5% discontinued contraceptive use by follow-up. Multinomial logistic regression, adjusting for women's characteristics, indicated the relative risk of contraceptive discontinuation doubled when women did not discuss pregnancy avoidance with their partner prior to contraceptive use. Partner influence was not related to method switching. Among non-users at baseline, partner support for future contraceptive use was associated with nearly three-fold increased odds of contraceptive adoption.

## Significance

These results highlight the potentially supportive role of male partners in contraceptive adoption. Future research is encouraged to elucidate the complex pathways between couple-based decision-making and contraceptive dynamics through further prospective studies.

## Introduction

With rapid increases in contraceptive coverage, there is growing interest in examining contraceptive use dynamics, particularly discontinuation, which increasingly contributes to unmet need for family planning. Global estimates using the Demographic and Health Survey (DHS) data from 34 low- and middle-income countries (LMICs) suggest that 38% of unmet need for contraception is attributable to the discontinuation of contraceptive methods while still stating a desire to avoid pregnancy [1]. Additionally, contraceptive discontinuation is estimated to account for one-third of unintended pregnancies [2]. In LMICs, contraceptive discontinuation is relatively common [3, 4], and varies by a range of factors, including method type, user characteristics and circumstances, and the quality of contraceptive counseling [5–7]. Similar factors also shape contraceptive adoption [8, 9], which importantly addresses access to contraception and method satisfaction of new users.

To date, the majority of research on determinants of contraceptive dynamics has focused on women's characteristics. Less attention has been given to partner-related influences, despite theoretical justification for their investigation. For example, Miller et al. propose that each partner's fertility intentions are influenced by individual desires, as well as the perceived desires of the other partner, and both may be continually moderated by spousal communication [10]. They posit that in cases where there is disagreement, the weight of each partner may not be equal in deciding the final reproductive behavior [10]. As such, most research on partner dynamics has primarily examined partners as barriers to contraceptive use, using rationales such as the Theory of Gender and Power, which specifies the potential impact of gender inequities across labor, power, and relationship domains [11]; these inequities have a cascading impact on women's health, including their access to and use of contraception [12, 13]. Studies in the United States, and more recently within LMICs, have examined partner-perpetrated reproductive coercion as influencing contraceptive non-use and subsequent unintended pregnancy [14–16]. This research has generally not been framed to examine a more extensive range of partner roles, including the potential influence of partner support on contraceptive dynamics, though this substantive focus is growing [17–19].

Moreover, quantitative studies examining partner influence on women's contraceptive use in LMICs tend to conflict with qualitative narratives, further obscuring this relationship. Few

nationally representative quantitative studies have examined partner influence outside of cross-sectional surveys, such as the DHS, which indicate that partner opposition is an infrequently reported reason for non-use of contraception. For example, among women with an unmet need for contraception, the percentage of women who reported that their primary reason for non-use was partner opposition was only 3.8% in Latin American and the Caribbean, 11.2% in Asia, and 9.5% in Africa [20]. Similar to reasons for contraceptive non-use, few women report discontinuation of contraceptive methods due to husband opposition. Among recent DHS surveys in sub-Saharan Africa, discontinuation due to husband opposition ranged from 1% in Ethiopia (2016) to 9.9% in Guinea (2018) [21]. Despite these quantitative findings, a growing evidence base from qualitative studies conducted in LMICs has focused on how partners may inhibit women's contraceptive use [17, 22–26]. These studies indicate women's preference to use contraception covertly if faced with partner opposition, however, they have mostly explored partners as barriers to contraceptive use and do not probe more extensively on experiences with partners as potential enablers to using contraception.

The different perspectives offered from large-scale quantitative surveys relative to more specific qualitative research raise a number of concerns regarding measurement and conceptualization of partner influence in large-scale surveys. Some studies suggest that partner opposition may be underreported, thereby attenuating its measured impact on contraceptive use [18,19]. Large-scale surveys, including the DHS, compute discontinuation rates using only the primary reason for discontinuation; this analytic approach may reduce the proportion of discontinuation due to husband opposition, as many women also report desires to get pregnant or concerns about side-effects/health concerns as their primary reason for discontinuation.

Beyond measurement issues, the widespread focus on partner opposition, while critical to address from a human rights and public health perspective, only captures one end of a spectrum of partner influences; this narrow scope omits the potential for examination of positive partner influences on women's adoption and continuation of contraception. Partner support is not just the lack of opposition—recent cross-sectional studies have examined the positive roles male partners may play in contraceptive decision-making. One study among pregnant women in rural Ghana found that perceived partner acceptability of family planning was one of the strongest predictors of the intention to use family planning in the future [27]. Another study of pregnant women and their partners in southeast Nigeria showed that men who stated support for their spouses' use of contraception were more likely to have spouses who intended to use contraception in the future [28]. Both studies highlight the central role of partner support in contraceptive decision-making, however they examine *intent* to use contraception, rather than contraceptive use. Qualitative data from South Africa have contextualized findings of quantitative studies by highlighting the supportive roles men play in facilitating access to family planning clinics, initiating contraceptive use, or improving adherence to methods and scheduled visits [17]. To date, only one longitudinal study conducted in Ghana linked spousal communication prospectively with contraceptive use, yet this study was not designed to be nationally representative [29].

Given the importance of gender and partner dynamics in reproductive decision-making, this study seeks to understand the predictive effect of partner influences, including partner support for contraception, discussions with partners about avoiding pregnancy prior to use, and perceived partner fertility intentions, on women's contraceptive use dynamics (adoption, discontinuation, and switching) over a one-year period in a nationally representative sample in Uganda. As programs continue to expand the emphasis on male engagement in family planning services, understanding the role of partner support in contraceptive dynamics will be crucial. Messages and interventions should be tailored to specific contraceptive dynamics to focus not only on addressing contraceptive discontinuation, but also on increasing adoption and continuation.

## Methods

### Study design

Longitudinal data for this analysis came from Performance Monitoring and Accountability 2020's (PMA2020) Uganda Round 6 survey (herein referred to as baseline), fielded in April-May 2018, with follow-up data from the Round 6 Follow-up Study (herein referred to as follow-up) conducted in May-June 2019. PMA2020 is a nationally representative, multi-stage, cluster survey of women aged 15–49. At baseline, 110 enumeration areas (EAs; geographical areas of approximately 200 households defined by the census), were selected using probability proportional-to-size sampling; all occupied households in the selected areas were enumerated. Forty-four households were randomly selected within each EA, and after completing a household survey, all women age 15–49 who were either usual members of the selected households or who slept in the household the night prior were approached for interview. After eligibility was confirmed, women were asked to complete written consent; if the woman agreed to participate, she was then interviewed by a trained resident interviewer. For unmarried women aged 15–17, written consent was first obtained from a parent or legal guardian prior to the women providing written assent to participate. Further information on the methodological design of PMA2020 cross-sectional surveys is available from www.pmadata.org and Zimmerman et al. [30].

A total sample of 4,288 women were interviewed at baseline. At time of interview, women were asked to consent for follow up, of which 4,095 agreed (95.4%). A total of 2,755 women were re-interviewed at follow-up (follow-up rate = 67.2%; Fig 1).

Baseline face-to-face interviews collected a range of information on women's sociodemographic characteristics, reproductive history, fertility status and intentions, and contraceptive behaviors. Women were also asked about their partners' fertility intentions and support of family planning. The follow-up survey focused on pregnancy intentions and outcomes, and contraceptive behaviors since the baseline interview.

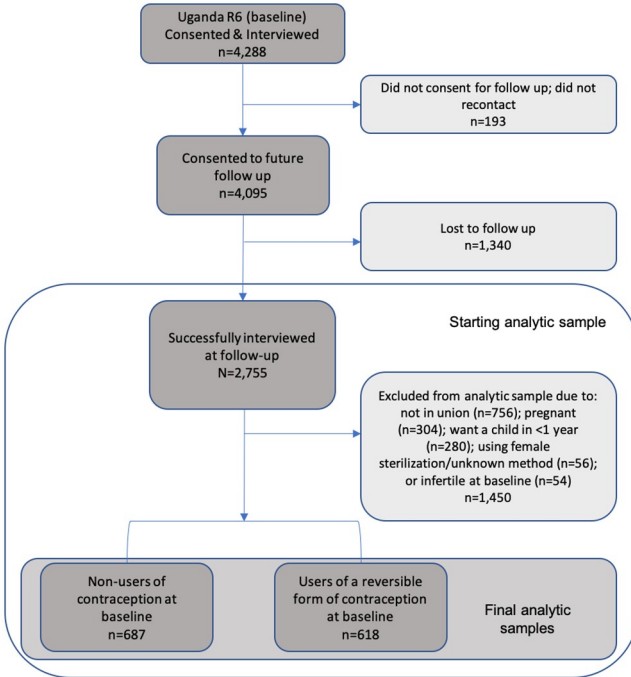

**Fig 1. Analytic sub-samples.**

All data collection procedures were approved by Institutional Review Boards (IRBs) at the Makerere University School of Public Health (HDREC 081) and Uganda National Council for Science and Technology (SS3400) in Kampala, Uganda, and the Johns Hopkins Bloomberg School of Public Health in Baltimore, USA.

## Analytic samples

Fig 1 shows a flow chart of the inclusion and exclusion criteria used to produce the two analytic samples included in this study. Both samples selected women who completed baseline and follow up-surveys and were in union at baseline; samples excluded women who did not have a partner or husband (n = 756). Women were also excluded if, at baseline, they were pregnant (n = 304), sterilized (n = 56), or indicated they were infertile (n = 54). In addition, women who reported wanting a child in the next year were excluded, as they would not be in need of contraception (n = 280). The analytic sample for the contraceptive discontinuation/ switching analysis included 618 users of reversible contraception, including traditional method users, at baseline. The analytic sample for the contraceptive adoption analysis included 687 women who were not using contraception at baseline.

## Loss-to-follow-up weights

Due to potential bias from loss-to-follow-up, an inverse propensity score was constructed by estimating a multivariate regression model with women's age, parity, marital status, schooling, wealth quintile, and residence as covariates. The predicted probability of loss-to-follow-up was then multiplied by the original baseline individual weight and its inverse was applied to re-weight the follow-up responses appropriately. S1 Table compares the responses of the full sample of women at baseline and those that completed follow-up, both weighted and unweighted. While there was differential loss-to-follow-up across sociodemographic characteristics, as well as our independent variables of interest, after adjusting for the predicted probability of loss-to-follow-up, there were no differences in the baseline characteristics of all women in the baseline versus those who were followed-up one year later. These sensitivity analyses indicate that the weighted findings in this analysis can be interpreted to reflect the national characteristics of women of reproductive age for baseline year.

## Measures

Primary dependent variables focused on contraceptive dynamics, namely contraceptive continuation, discontinuation, switching, and adoption (defined subsequently). All contraceptive dynamics measures were assessed by comparing women's contraceptive use status at baseline and follow-up interviews. Contraceptive use status was defined at each survey as the response to the question "*Are you or your partner currently doing something or using any method to delay or avoid getting pregnant*?" followed by a list of methods, if affirmative.

- *Contraceptive continuation/ discontinuation/ switching*: Contraceptive continuation, discontinuation, and switching were assessed only among women who indicated contraceptive use at baseline. Women were defined as "continuers" if they were using the same method at baseline and follow-up, defined as "switchers" if they were using a different method at follow-up than what they reported at baseline, and as "discontinuers" if they reported non-use at follow-up.

- *Contraceptive adoption*: Contraceptive adoption was assessed among non-users of contraception at baseline. Women were categorized as "adopters" if they reported using at follow-up or as "continued non-users" if they reported non-use again at follow-up.

## Independent variables

Three independent variables of interest were examined separately. All partner characteristics reported were as perceived by the female partner.

*Partner support for contraceptive use* was assessed using two questions asked at baseline. To assess support of current or future contraceptive use, users of contraception at baseline were asked if their partner was supportive of their contraceptive use; non-users at baseline were asked if their partner would be supportive of them using contraception in the future. Response categories included: yes, no, and don't know. Women who responded "don't know" to current contraceptive use support (n = 6) were excluded from regression analyses.

*Women's discussions with their partners surrounding the decision to avoid a pregnancy prior to contraceptive use* were examined among all women who were using contraception at baseline. Women were asked if they discussed the decision to avoid a pregnancy with their partner prior to starting their contraceptive method (binary response: yes/no).

*Partner's perceived fertility intentions* were assessed at baseline for all women via a single item: *"Does your husband/partner want to have a/another child with you within two years?"* Response categories included: yes, no, and unsure.

## Covariates

Sociodemographic covariates at baseline were considered, including women's age (15–24, 25–34, 35+ years), education (none, primary, secondary+), parity (0–2, 3–4, 5+ children), polygynous relationship, household wealth (quintiles), and residence (urban/rural), which have been shown to be related to contraceptive use. All covariates were modelled as binary or categorical, and small categories were combined given baseline distributions. For parity, nulliparous women were grouped with women who had 1–2 children, given skewedness towards high childbearing.

## Analytical approach

First, exploratory analyses assessed the distributions of sociodemographic characteristics and independent variables measuring partner influence (partner support, discussions, and fertility intentions) among each analytic sample.

Second, bivariate associations were conducted among contraceptive users at baseline, showing the percent of women who continued use, switched methods, or who had discontinued at follow-up, across socio-demographic characteristics and partner influences; design-based F statistics were used to detect significant differences between contraceptive continuers, switchers, and discontinuers. The same approach was applied to the study of contraceptive adoption, by examining associations between partner influences, sociodemographic characteristics, and contraceptive adoption at follow-up among women who were not using contraception at baseline. Sensitivity analyses adjusted for method longevity (long vs. short-acting method); however, no significant differences were observed (results not shown).

Among women who were using contraception at baseline, bivariate and multivariable multinomial logistic regressions were conducted to examine the influence of partner fertility intentions and support for contraception, and discussions with a partner to avoid a pregnancy, on contraceptive switching and discontinuation. Adjusted analyses accounted for women's age, schooling level, parity, polygyny, urban/rural residence, and wealth quintile. Among women who were not using contraception at baseline, bivariate and multivariable logistic regression models estimated the odds ratios of contraceptive adoption at follow-up, according to partner support for future use of contraception and adjusting for the same baseline sociodemographic characteristics.

All analyses were conducted using Stata 16 (College Station, TX), accounting for within-EA clustering effects and multi-stage stratified cluster survey design of PMA, in addition to adjustment for differential loss-to-follow-up using inverse propensity score weights. P-values of <0.05 were considered statistically significant; given sample size limitations, p-values <0.10 were considered marginally significant.

## Results

Table 1 describes the characteristics of the two analytic samples at baseline. In terms of contraceptive users at baseline, over two-thirds of women were younger than 35 years (70.4%), almost all had at least a primary school education (93.4%), and the majority lived in rural areas (80.7%). Number of children was evenly split across the three categories: 31%

**Table 1. Descriptive characteristics of contraceptive users and non-users at baseline (n = 1,305).**

| Baseline characteristics | | Contraceptive status at baseline (%) | |
| --- | --- | --- | --- |
| | | Contraceptive users at baseline (n = 618) | Contraceptive non-users at baseline (n = 687) |
| Age | 15–24 years | 27.2 | 23.7 |
| | 25–35 years | 43.2 | 38.6 |
| | 35 plus years | 29.6 | 37.7 |
| Highest schooling level | None | 6.6 | 17.3 |
| | Primary | 56.2 | 61.7 |
| | Secondary or higher | 37.2 | 21.1 |
| Parity | 0–2 children | 30.9 | 26.3 |
| | 3–4 children | 31.9 | 25.4 |
| | 5 plus children | 37.2 | 48.3 |
| Partner has other wives | No | 66.1 | 63.6 |
| | Yes | 26.4 | 32.6 |
| | Don't know | 7.6 | 3.9 |
| Household wealth category | Lowest | 18.2 | 32.0 |
| | Middle lowest | 16.1 | 23.9 |
| | Middle | 22.0 | 18.3 |
| | Middle higher | 21.8 | 12.4 |
| | Highest | 21.9 | 13.5 |
| Urban residence | Yes | 19.3 | 17.6 |
| Type of method used | Modern | 86.8 | - - |
| | Traditional | 13.3 | - - |
| Partner support for current contraceptive use | No Support | 15.7 | - - |
| | Support | 83.4 | - - |
| | Don't know | 0.9 | - - |
| Discussed decision to avoid pregnancy with partner before method | No | 17.9 | - - |
| | Yes | 82.1 | - - |
| Partner support for future contraceptive use | No support for future use | - - | 37.0 |
| | Support for future use | - - | 54.8 |
| | Don't know | - - | 8.2 |
| Partner fertility intentions | Partner wants child within 2 years | 39.9 | 43.5 |
| | Partner doesn't want child within 2 years | 42.5 | 38.0 |
| | Don't know | 17.6 | 18.5 |

of women reported having 0–2 children, 32% reported 3–4 children, while 37% reported having 5 or more children. Two-thirds of women stated their partners had no other wives. Most contraceptive users at baseline reported using a modem method of contraception (86.8%), stated that their partners supported their use of contraception (83.4%) and that they had discussed the decision to avoid a pregnancy with their partners before starting their current method (82.1%). Approximately 40% of contraceptive users reported that their partners wanted a child within two years and another 42% believed their partners didn't want a child within two years; the remaining women stated they did not know their partners' fertility intentions (17.6%).

The characteristics of the non-users at baseline were slightly different (Table 1). Almost two-thirds of the sample were younger than 35 years (62.3%), and while a majority reported at least a primary school education (82.7%), 17.3% had no formal education. Almost half of the non-users at baseline had five or more living children (48.3%). Almost two-thirds of women reported their partners had no other wives (63.6%). Slightly more than half of women stated their partner would be supportive of future contraceptive use (54.8%), while 8.2% were not sure. Over forty percent of users reported that their partners wanted a child within two years, while 38.0% indicated that their partners didn't want a child within two years; the remaining women were unsure of their partners' fertility intentions (18.5%).

Table 2 shows the bivariate associations between baseline characteristics and contraceptive use dynamics at follow-up among users at baseline. Overall, 39.1% of women had continued their contraceptive method at follow-up, while 30% had switched methods and 31% had discontinued completely. Parity was the only sociodemographic characteristic significantly associated with switching or discontinuation; women with five or more children showed the highest proportion of continuation (43.5%), while women with fewer children had the highest proportion of discontinuation (40.0%; p = 0.02). Engaging in discussions about avoiding pregnancy prior to initiating contraceptive use was marginally associated with contraceptive dynamics (p = 0.07).

In the adjusted analysis (Table 3), none of the partner influences nor perceptions of partner fertility intentions were related to switching patterns. On the other hand, women who reported having contraceptive use support from their partners had a reduced risk of discontinuing, relative to women who said their partners did not support their contraceptive use, though this association was only marginally significant (aOR = 0.59, p = 0.08). Similarly, participating in discussions about pregnancy avoidance prior to adopting the contraceptive method was significantly associated with a lower risk of discontinuation, relative to women who did not have these discussions with their partners (aOR = 0.55, p = 0.04). Partners' perceived fertility intentions were not related to either switching or discontinuation.

Factors informing contraceptive adoption are displayed in Table 4. We found that 32.1% of non-users of contraception at baseline had adopted contraception by the follow-up survey. Younger women (age 15–34) had higher proportions of adoption than those aged 35 and above. Similarly, adoption increased with increasing education level. Further, 42.8% of women who reported their partners would support future contraceptive use had adopted a method by follow-up, whereas only 19.5% of those without reported support had started using a method (p<0.001).

In the adjusted logistic regression analysis (Table 5), partner support for future contraceptive use was associated with 2.74 increased odds of adoption compared to those who did not perceive partner support in future contraceptive use (p<0.05). Further, women who reported that their partner did not want a child within two years were more likely to adopt contraception than those who reported that their partner wanted a child soon, although the association did not reach statistical significance (aOR = 1.61, p = 0.09).

**Table 2. Bivariate associations between baseline characteristics and contraceptive status at follow-up, among baseline contraceptive users (n = 618).**

| Baseline characteristics | | Row n | Contraceptive status at follow-up | | | p-value |
|---|---|---|---|---|---|---|
| | | | Women who continued | Women who switched methods | Women who discontinued | |
| | | | % | | | |
| Total n (%) | | 618 | 39.1 | 30.3 | 30.6 | |
| Age | 15–24 years | 136 | 35.4 | 28.0 | 36.7 | 0.22 |
| | 25–35 years | 274 | 36.3 | 35.0 | 28.7 | |
| | 35 plus years | 208 | 46.7 | 25.6 | 27.7 | |
| Highest schooling level | None | 53 | 51.8 | 9.1 | 39.1 | 0.07 |
| | Primary | 376 | 39.4 | 28.5 | 32.1 | |
| | Secondary or higher | 189 | 36.5 | 36.8 | 26.7 | |
| Parity | 0–2 children | 157 | 28.3 | 31.7 | 40.0 | **0.02** |
| | 3–4 children | 195 | 44.4 | 32.6 | 23.0 | |
| | 5 plus children | 266 | 43.5 | 27.2 | 29.3 | |
| Partner has other wives | No | 410 | 40.3 | 30.5 | 29.1 | 0.94 |
| | Yes | 167 | 36.5 | 30.0 | 33.5 | |
| | Don't know | 41 | 37.3 | 29.5 | 33.2 | |
| Household wealth category | Lowest | 134 | 40.7 | 25.7 | 33.6 | 0.43 |
| | Middle lowest | 113 | 33.8 | 28.3 | 37.9 | |
| | Middle | 143 | 34.2 | 32.7 | 33.1 | |
| | Middle higher | 108 | 41.4 | 36.7 | 21.9 | |
| | Highest | 120 | 44.3 | 26.8 | 28.8 | |
| Residence | Urban | 122 | 37.1 | 39.3 | 23.6 | 0.27 |
| | Rural | 496 | 39.6 | 28.2 | 32.3 | |
| Type of method used | Modern | 537 | 39.0 | 30.8 | 30.2 | 0.85 |
| | Traditional | 81 | 40.0 | 27.1 | 32.9 | |
| Partner support for current contraceptive use | No Support | 95 | 36.7 | 23.1 | 40.2 | 0.32 |
| | Support | 517 | 39.5 | 31.8 | 28.7 | |
| Discussed decision to avoid pregnancy with partner before method | No | 107 | 35.0 | 23.3 | 41.8 | 0.07 |
| | Yes | 510 | 39.9 | 31.9 | 28.2 | |
| Partner fertility intentions | Partner wants child within 2 years | 244 | 38.0 | 29.8 | 32.1 | 0.89 |
| | Partner doesn't want child within 2 years | 265 | 41.5 | 29.7 | 28.8 | |
| | Don't know | 108 | 35.9 | 32.5 | 31.5 | |

Boldface indicates p<0.05 from design-based F-statistic.

## Discussion

Our findings reflect the multi-faceted ways in which partners can influence contraceptive use and provide greater context to the role that partners play in contraceptive adoption, switching, and discontinuation. Specifically, we found that partner support was strongly associated with contraceptive adoption among non-users at baseline, after accounting for demographic characteristics. Among women who were using contraception at baseline, findings suggest that open communication about one's desire to prevent pregnancy prior to using contraception may support contraceptive continuation. While women's perceptions of their partners' fertility intentions were not associated with continued use of contraception, they did appear to marginally affect contraception adoption.

**Table 3. Bivariate and multivariable multinomial regression of contraceptive switching and discontinuation relative to continued use by partner influence, among baseline contraceptive users (n = 618).**

| | Switched | | Discontinued | |
|---|---|---|---|---|
| | Unadjusted | Adjusted≥ | Unadjusted | Adjusted≥ |
| | RRR (95% CI) | | | |
| Partner support for current contraceptive use | | | | |
| No support | ref | ref | ref | ref |
| Support | 1.28 (0.60–2.71) | 1.59 (0.73–3.49) | 0.66 (0.39–1.13) | 0.59∞ (0.33–1.05) |
| Discussed decision to avoid pregnancy with partner | | | | |
| No discussion | ref | ref | ref | ref |
| Discussion | 1.20 (0.58–2.49) | 1.29 (0.63–2.68) | **0.59* (0.35–0.99)** | **0.55* (0.32–0.96)** |
| Partner fertility intentions | | | | |
| Wants a child < 2 years | ref | ref | ref | ref |
| Doesn't want a child < 2 years | 0.91 (0.55–1.50) | 0.99 (0.54–1.83) | 0.82 (0.45–1.49) | 0.89 (0.50–1.57) |
| Don't know | 1.16 (0.61–2.18) | 1.27 (0.63–2.54) | 1.04 (0.54–2.00) | 1.16 (0.59–2.28) |

≥Adjusted for age, parity, education, polygyny, residence, and wealth quintile.

∞p<0.10;

*p<0.05;

**p<0.01;

***p<0.001; Boldface indicates p<0.05.

Taken together, these results highlight the important and positive role that partners may play in contraceptive use dynamics. Involving partners in discussions around fertility decisions to delay or limit childbearing may be important components for interventions and programs that aim to improve contraceptive continuation. While these findings are fairly intuitive, few longitudinal studies have prospectively examined these relationships. For example, increased spousal communication around family planning has been linked with higher contraceptive use and lower unmet need [31], however, directionality of this relationship remained unclear. Establishing the temporal relationship between partner influence and contraceptive discontinuation, as demonstrated by the present study, confirms the predictive effect of spousal communication on a nationally representative level in Uganda. Further, these results offer evidence to support interventions that aim to improve communication within couples around childbearing decisions and family planning. Specifically, interventions should aim to combat harmful social norms that may label conversations around contraception as taboo within a couple dyad.

The marginal results on partner support and contraceptive discontinuation and null findings for partner support and switching may suggest that partner influence holds less weight once a woman is already using contraception. Qualitative research among women who use contraception covertly due to partner opposition indicates high motivation for contraceptive use to avoid disclosure [23, 24]. Further, research in both high- and low-income settings indicates that motivations are highly predictive of contraceptive continuation [32, 33]. In the absence of information on the strength of women's motivations surrounding pregnancy prevention, this analysis could not disentangle partner support from women's own motivations. Continued research is needed to understand women's nuanced decision-making practices and their motivations in adopting and continuing contraceptive use, particularly for women who may be unable to garner initial partner support.

This study has several limitations. First, only 67% of the original baseline sample was retained at follow-up, mostly due to loss of women at the household level. While the loss-to-

**Table 4. Bivariate associations between baseline characteristics and contraceptive status at follow-up, among contraceptive non-users at baseline (n = 687).**

| Baseline characteristics | | n | Continued non-user | Adopters | p-value |
|---|---|---|---|---|---|
| | | | % | | |
| Total n (%) | | 687 | 67.9 | 32.1 | |
| Age | 15–24 years | 137 | 60.6 | 39.4 | **<0.001** |
| | 25–35 years | 253 | 61.5 | 38.5 | |
| | 35 plus years | 297 | 79.0 | 21.0 | |
| Highest schooling level | None | 162 | 81.4 | 18.6 | **0.02** |
| | Primary | 413 | 68.2 | 31.8 | |
| | Secondary or higher | 112 | 56.0 | 44.0 | |
| Parity | 0–2 children | 160 | 68.7 | 31.3 | **0.03** |
| | 3–4 children | 170 | 59.0 | 41.0 | |
| | 5 plus children | 357 | 72.1 | 27.9 | |
| Partner has other wives | No | 413 | 65.3 | 34.7 | 0.46 |
| | Yes | 252 | 73.3 | 26.7 | |
| | Don't know | 22 | 64.9 | 35.1 | |
| Household wealth category | Lowest | 268 | 76.2 | 23.8 | 0.11 |
| | Middle lowest | 158 | 69.9 | 30.1 | |
| | Middle | 120 | 67.8 | 32.2 | |
| | Middle higher | 76 | 57.4 | 42.6 | |
| | Highest | 65 | 54.2 | 45.8 | |
| Residence | Urban | 108 | 63.1 | 36.9 | 0.51 |
| | Rural | 579 | 68.9 | 31.1 | |
| Partner support for future contraceptive use | No support for future use | 277 | 80.5 | 19.5 | **<0.001** |
| | Support for future use | 338 | 57.2 | 42.8 | |
| | Don't know | 64 | 81.9 | 18.1 | |
| Partner fertility intentions | Partner wants child < 2 years | 293 | 69.0 | 31.0 | 0.60 |
| | Partner doesn't want child < 2 years | 251 | 65.2 | 34.8 | |
| | Don't know | 141 | 70.4 | 29.6 | |

Boldface indicates p<0.05 from design-based F-statistic.

follow-up was differential by expected socio-demographic characteristics (S1 Table), loss-to-follow-up weighting adjustment was able to restore the panel composition to that of the original sample. However, underlying differences between the samples could persist. Further, given low contraceptive prevalence in Uganda, the high loss-to-follow-up observed in our sample restricted the sample sizes for examining contraceptive use dynamics and potentially limited our ability to detect statistically meaningful differences between adopters, discontinuers, and switchers. Moreover, these analyses were limited by measurement issues; as indicated in previous research [32–34], the timeframe for partner fertility intentions (two years) did not match the same timeframe as a woman's own fertility intentions (one year). Lastly, all partner characteristics were reported by the female respondent, rather than by her partner. Few studies have evaluated the accuracy of proxy-partner reporting, especially around family planning and fertility measures, in sub-Saharan African contexts [35–38].

Despite these limitations, this study adds to a small, but growing, body of longitudinal evidence on partner influence and contraceptive use dynamics and has meaningful implications for family planning programs and providers. Foremost, interventions should be targeted at both couples and women alone in order to increase uptake of contraception among those seeking it, reduce discontinuation among those still in need, and promote switching where it is

**Table 5. Bivariate and multivariable logistic regression of contraceptive adoption relative to continued non-use by partner influence, among baseline contraceptive non-users (n = 687).**

| | Unadjusted | Adjusted$^{\geq}$ |
|---|---|---|
| | OR (CI) | |
| Partner support for future contraceptive use | | |
| No support | ref | ref |
| Support | **3.10*** (1.86–5.16)** | **2.74*** (1.48–5.09)** |
| Don't know | 0.91 (0.33–2.52) | 1.15 (0.36–3.62) |
| Partner's fertility intentions | | |
| Wants a child within 2 years | ref | ref |
| Doesn't want a child within 2 years | 1.19 (0.75–1.88) | *1.61$^{\infty}$ (0.93–2.78)* |
| Don't know | 0.94 (0.58–1.50) | 1.39 (0.82–2.35) |

$^{\geq}$Adjusted for age, parity, education, polygyny, residence, and wealth quintile.

$^{\infty}$p<0.10;

$^{*}$p<0.05;

$^{**}$p<0.01;

$^{***}$p<0.001; Boldface indicates p<0.05.

sought. These results highlight the benefits of engaging male partners in family planning programs, particularly to facilitate adoption among non-users who want to delay or limit childbearing, as evidenced by decreased adoption among those without partner support (20% vs 43%). Programs that support spousal communication around family planning, avoiding or planning for a pregnancy, and fertility intentions could be key in achieving reproductive goals [39–41].

While this study provides some evidence of male partners' supporting roles, the specifics remain poorly understood. Future research must include men in studies surrounding fertility preferences and views on contraceptive use, rather than assessing solely by women's reports. Further, this research should seek to understand characteristics of partners who may be supportive of contraceptive use, including individual, couple, family, and contextual factors that promote contraceptive discussions and use. Lastly, a better understanding of the spectrum from partner opposition to support is needed, including reasons for support, non-support, and opposition. Future research will require improved measurement approaches to understand the range of roles that partners play in contraceptive dynamics to further disentangle pathways that may either help or hinder contraceptive use.

## Conclusions

Contraceptive use is often a couple-based decision, and while reproductive coercion gravely infringes on women's health and rights, this research also points to the supportive role of partners in contraceptive decisions. Future research is encouraged to consider the spectrum of partner roles in contraceptive decisions and the factors promoting male positive engagement. While simultaneously ensuring that women's rights and bodily autonomy are protected, increased male engagement in reproductive health programs may facilitate improved knowledge and healthier couple communication.

## Supporting information

**S1 Table. Comparison of baseline and follow-up samples with and without loss-to-follow-up weighting.**
(DOCX)

## Author Contributions

**Conceptualization:** Dana O. Sarnak, Shannon N. Wood, Celia Karp, Fredrick Makumbi, Caroline Moreau.

**Data curation:** Fredrick Makumbi, Simon P. S. Kibira.

**Formal analysis:** Dana O. Sarnak.

**Investigation:** Linnea A. Zimmerman, Fredrick Makumbi, Simon P. S. Kibira.

**Methodology:** Dana O. Sarnak, Shannon N. Wood, Linnea A. Zimmerman, Caroline Moreau.

**Project administration:** Dana O. Sarnak, Linnea A. Zimmerman, Fredrick Makumbi, Simon P. S. Kibira.

**Supervision:** Linnea A. Zimmerman.

**Writing – original draft:** Dana O. Sarnak, Shannon N. Wood.

**Writing – review & editing:** Dana O. Sarnak, Shannon N. Wood, Linnea A. Zimmerman, Celia Karp, Fredrick Makumbi, Simon P. S. Kibira, Caroline Moreau.

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
