## [Decision Letter · Decision Letter 0]

9 Nov 2020

PONE-D-20-26947

The role of partner influence in contraceptive adoption, discontinuation, and switching in a nationally representative cohort of Ugandan women

PLOS ONE

Dear Dr. Sarnak,

Thank you for submitting your manuscript to PLOS ONE. After careful consideration, we feel that it has merit but does not fully meet PLOS ONE’s publication criteria as it currently stands. Therefore, we invite you to submit a revised version of the manuscript that addresses the points raised during the review process.

We look forward to receiving your revised manuscript.

Kind regards,

Kannan Navaneetham, PhD

Academic Editor

PLOS ONE

Journal Requirements:

2. Please include information about IRB approval in the manuscript as well.

3.We note that you have indicated that data from this study are available upon request. PLOS only allows data to be available upon request if there are legal or ethical restrictions on sharing data publicly. For information on unacceptable data access restrictions, please see http://journals.plos.org/plosone/s/data-availability#loc-unacceptable-data-access-restrictions.

Reviewers' comments:

Reviewer's Responses to Questions

**Comments to the Author**

1. Is the manuscript technically sound, and do the data support the conclusions?

Reviewer #1: Yes

2. Has the statistical analysis been performed appropriately and rigorously? 

Reviewer #1: Yes

3. Have the authors made all data underlying the findings in their manuscript fully available?

Reviewer #1: Yes

4. Is the manuscript presented in an intelligible fashion and written in standard English?

Reviewer #1: Yes

5. Review Comments to the Author

Reviewer #1: This paper adds to our further understanding about the important role that male partners play in reproductive health and contraceptive use. Although the findings are not surprising or new they do contribute to the literature - especially for those women who are non-contraceptive users.

I do have a few comments:

I would recommend for the authors to expand their literature review a bit.

Line 67-71 - Quite a bit of work (both quantitative and qualitative) is being done in the sub-Saharan setting about the involvement of male partners in contraceptive use. So I am not convinced that this statement holds very true.

Line 78-79 has a grammatical error. I think the word "use" is missing.

Line 78-81 - Please review the cited literature again. Some of the studies described in this paragraph actually does focus on the positive influence that male partners have on contraceptive use. Also, not all were framed to explore the male partner's influence as a barrier.

Analytical and Results section

I think the statistical and analytical section was well described and supported the findings.

One aspect that remains key in studies that examine male partner influence on contraceptive use is the fact that men are so often excluded from these studies. There is quite a large body of literature reporting women's perspectives on male involvement in contraceptive use - while the male perspective, or couple perspective, would add so much more clarity and understanding but is lacking. I do feel that this is where there is a silent gap in the DHS/large survey data and not really on the effect of the male partner on contraceptive use as reported by the female partner.

It is a pity about the large loss to follow-up rate.

Discussion

Only a small number of women (32.1%) started using contraception over the year follow up period. It is quite significant that such a small percentage of those women started contraceptive use without male partner support - which indicates just how important male partner support is in that setting.

Overall the findings were not surprising. It is well documented that communication plays a key role in contraceptive uptake and positive partner involvement. The results presented here support that. Probably the most interesting finding was that male partner influence seem to wane with increased contraceptive use by the female partner. This area needs further exploration, especially in the sub-Saharan region where patriarchal views still dominate. It is a bit of a concern that individual motivation could not be separated from partner influence.

Overall

I think this is a well written article with some important findings that adds to our understanding about male influence on contraceptive use.

6. PLOS authors have the option to publish the peer review history of their article (what does this mean?). If published, this will include your full peer review and any attached files.

Reviewer #1: **Yes: **Yolandie Kriel

---

## [Author Response · Author response to Decision Letter 0]

8 Dec 2020

Journal Requirements:

2. Please include information about IRB approval in the manuscript as well.

We have the following text regarding IRB approval in the manuscript on page 7, line 151-154: “All data collection procedures were approved by Institutional Review Boards (IRBs) at the Makerere University School of Public Health (HDREC 081) and Uganda National Council for Science and Technology (SS3400) in Kampala, Uganda, and the Johns Hopkins Bloomberg School of Public Health in Baltimore, USA.” Please let us know if additional information is needed. 

3.We note that you have indicated that data from this study are available upon request. PLOS only allows data to be available upon request if there are legal or ethical restrictions on sharing data publicly. For information on unacceptable data access restrictions, please see http://journals.plos.org/plosone/s/data-availability#loc-unacceptable-data-access-restrictions.

Apologies for the confusion. PMA uses the same model as other publicly available datasets such as DHS; de-identified data is accessible to the public from https://www.pmadata.org/data/available-datasets through a registration process. We revised the language on the submission portal to: “Data for this study are publicly available at https://www.pmadata.org/data/available-datasets. Data are free to download and use; users are required to register for a PMA dataset account. The specific datasets used in this study are the Uganda Round 6 (2018) Household & Female survey and the Uganda Round 6 Follow-up (2019) Household & Female survey. The DOI for Uganda Round 6 (“baseline”) is doi: 10.34976/D41K-SH61 and the DOI for Uganda Round 6 Follow Up (“follow-up”) is doi: 10.34976/9x2b-nd72.”

There are no ethical or legal restrictions on sharing the de-identified dataset.

PMA provides data to the public via request at https://www.pmadata.org/data/available-datasets. This URL should suffice for the stable, public repository PLOS One requires. The DOI for Uganda Round 6 (“baseline”) is doi: 10.34976/D41K-SH61 and the DOI for Uganda Round 6 Follow Up (“follow-up”) is doi: 10.34976/9x2b-nd72.

We revised the text to read “results not shown.” We conducted a sensitivity analyses that included a covariate for method type (short acting vs long acting method). We decided against adding another table or supporting information files to show these results given that the inferences remained the same. The results we refer to come from the public access dataset that we refer to above. Please let us know if we misunderstood this requirement. 

We copied and pasted the abstract from the manuscript to the online submission form and they are now identical. 

We added the necessary captions for our supporting information file at the end of the manuscript, as described in the guidelines. 

Reviewers' comments:

Reviewer's Responses to Questions

Comments to the Author

1. Is the manuscript technically sound, and do the data support the conclusions?

Reviewer #1: Yes

2. Has the statistical analysis been performed appropriately and rigorously? 

Reviewer #1: Yes

3. Have the authors made all data underlying the findings in their manuscript fully available?

Reviewer #1: Yes

4. Is the manuscript presented in an intelligible fashion and written in standard English?

Reviewer #1: Yes

5. Review Comments to the Author

Reviewer #1: This paper adds to our further understanding about the important role that male partners play in reproductive health and contraceptive use. Although the findings are not surprising or new they do contribute to the literature - especially for those women who are non-contraceptive users.

I do have a few comments:

I would recommend for the authors to expand their literature review a bit.

Line 67-71 - Quite a bit of work (both quantitative and qualitative) is being done in the sub-Saharan setting about the involvement of male partners in contraceptive use. So I am not convinced that this statement holds very true.

Thank you for this suggestion. We have revised the introduction to include more of the literature on partner involvement and family planning in the region. Specifically, we have expanded the paragraph on positive influences that male partners can play in family planning on page 4, line 93. We highlight how male support can increase women’s intention to use family planning in the future, how male partners can facilitate access to facilities, adherence to methods, and how spousal communication has been linked prospectively with contraceptive use. We have also clarified the specific gap that our study fills in the research on page 5, line 111 : “Given the importance of gender and partner dynamics in reproductive decision-making, this study seeks to understand the predictive effect of partner influences, including partner’s support for contraception, discussions with partners about avoiding pregnancy prior to use, and perceived partner’s fertility intentions, on women’s contraceptive use dynamics (adoption, discontinuation, and switching) over a one-year period in a nationally representative sample in Uganda.”

Line 78-79 has a grammatical error. I think the word "use" is missing.

Thank you for catching this typo, we have updated the sentence with the word “use” on line 79.

Line 78-81 - Please review the cited literature again. Some of the studies described in this paragraph actually does focus on the positive influence that male partners have on contraceptive use. Also, not all were framed to explore the male partner's influence as a barrier.

Thank you for pointing this out. Upon revisiting the papers cited in that paragraph, we have revised the introduction to include a paragraph that focuses on the specific supportive roles male partners can play in family planning and contraception, lines 93-109.

Analytical and Results section

I think the statistical and analytical section was well described and supported the findings.

One aspect that remains key in studies that examine male partner influence on contraceptive use is the fact that men are so often excluded from these studies. There is quite a large body of literature reporting women's perspectives on male involvement in contraceptive use - while the male perspective, or couple perspective, would add so much more clarity and understanding but is lacking. I do feel that this is where there is a silent gap in the DHS/large survey data and not really on the effect of the male partner on contraceptive use as reported by the female partner.

We agree. One limitation of our study, which we mention in the discussion, is that our measures of male partners’ characteristics, e.g. their fertility intentions and their support for contraception, are reported by women respondents. Therefore, these proxy-reported indicators may differ from how male partners would respond to survey items. We conclude by calling for the inclusion of men in future research on contraceptive use and reproductive health more broadly. 

It is a pity about the large loss to follow-up rate.

We agree that the large loss-to-follow-up is not ideal; however, we feel that our loss-to-follow up weights did restore our panel sample to reflect the original sample in terms of key demographic variables. There could be unobserved differences and we state this as our primary limitation to the study. 

Discussion

Only a small number of women (32.1%) started using contraception over the year follow up period. It is quite significant that such a small percentage of those women started contraceptive use without male partner support - which indicates just how important male partner support is in that setting.

Thank you for this suggestion; we have called out the percentage difference in the percent of non-users that adopted with and without partner support in the discussion on line 389-390. 

Overall the findings were not surprising. It is well documented that communication plays a key role in contraceptive uptake and positive partner involvement. The results presented here support that. Probably the most interesting finding was that male partner influence seem to wane with increased contraceptive use by the female partner. This area needs further exploration, especially in the sub-Saharan region where patriarchal views still dominate. It is a bit of a concern that individual motivation could not be separated from partner influence.

We agree that there are nuances we were not able to capture in this study, e.g. comparing a woman’s own motivation to avoid pregnancy or use contraception to the influence of her partners. In an effort to address this, we restricted the sample of women to those who wanted to avert a pregnancy during the study period, so that they were all “at risk” of unintended pregnancy. Unfortunately, we do not have a way to measure the strength of their motivation to avoid pregnancy, and therefore cannot disentangle it from their reports of partner support. Further, this construct may not make sense among women who are currently users.

We also think that it is interesting that partner support was an important predictor of contraception adoption, yet current support of contraception was not predictive of discontinuation. We could not provide evidence-based explanations from our data, only postulations from other research. Further research should explore this in-depth as you suggest; specifically, identifying ways in which male partners can be supportive in adoption, continuation and switching, and whether and how these ways are similar or different. 

Overall

I think this is a well written article with some important findings that adds to our understanding about male influence on contraceptive use.

---

## [Editor Report · Decision Letter 1]

10 Dec 2020

The role of partner influence in contraceptive adoption, discontinuation, and switching in a nationally representative cohort of Ugandan women

PONE-D-20-26947R1

Dear Dr. Sarnak,

We’re pleased to inform you that your manuscript has been judged scientifically suitable for publication and will be formally accepted for publication once it meets all outstanding technical requirements.

Kind regards,

Kannan Navaneetham, PhD

Academic Editor

PLOS ONE
---

## [Editor Report · Acceptance letter]

2 Jan 2021

PONE-D-20-26947R1 

The role of partner influence in contraceptive adoption, discontinuation, and switching in a nationally representative cohort of Ugandan women 

Dear Dr. Sarnak:

I'm pleased to inform you that your manuscript has been deemed suitable for publication in PLOS ONE. Congratulations! Your manuscript is now with our production department. 

Kind regards, 

on behalf of

Professor Kannan Navaneetham 

Academic Editor

PLOS ONE